# Actigraph-Measured Movement Correlates of Attention-Deficit/Hyperactivity Disorder (ADHD) Symptoms in Young People with Tuberous Sclerosis Complex (TSC) with and without Intellectual Disability and Autism Spectrum Disorder (ASD)

**DOI:** 10.3390/brainsci10080491

**Published:** 2020-07-28

**Authors:** Tom Earnest, Elizabeth Shephard, Charlotte Tye, Fiona McEwen, Emma Woodhouse, Holan Liang, Fintan Sheerin, Patrick F. Bolton

**Affiliations:** 1Department of Child & Adolescent Psychiatry, Institute of Psychiatry, Psychology & Neuroscience, King’s College London, London SE5 8AF, UK; elizabeth.1.shephard@kcl.ac.uk (E.S.); charlotte.tye@kcl.ac.uk (C.T.); fiona.mcewen@kcl.ac.uk (F.M.); ho-lan.liang@gosh.nhs.uk (H.L.); fintansheerin@gmail.com (F.S.); patrick.bolton@kcl.ac.uk (P.F.B.); 2Social, Genetic & Developmental Psychiatry (SGDP) Centre, Institute of Psychiatry, Psychology & Neuroscience, King’s College London, London SE5 8AF, UK; 3Forensic and Neurodevelopmental Sciences, Institute of Psychiatry, Psychology & Neuroscience, King’s College London, London SE5 8AF, UK; emma.woodhouse@kcl.ac.uk; 4Great Ormond Street Hospital, London WC1N 3JH, UK

**Keywords:** tuberous sclerosis (TSC), attention-deficit/hyperactivity disorder (ADHD), activity levels, actigraphy, autism spectrum disorder (ASD), intellectual disability, epilepsy

## Abstract

Actigraphy, an objective measure of motor activity, reliably indexes increased movement levels in attention-deficit/hyperactivity disorder (ADHD) and may be useful for diagnosis and treatment-monitoring. However, actigraphy has not been examined in complex neurodevelopmental conditions. This study used actigraphy to objectively measure movement levels in individuals with a complex neurodevelopmental genetic disorder, tuberous sclerosis (TSC). Thirty participants with TSC (11–21 years, 20 females, IQ = 35–108) underwent brief (approximately 1 h) daytime actigraph assessment during two settings: movie viewing and cognitive testing. Multiple linear regressions were used to test associations between movement measurements and parent-rated ADHD symptoms. Correlations were used to examine associations between actigraph measures and parent-rated ADHD symptoms and other characteristics of TSC (symptoms of autism spectrum disorder (ASD), intellectual ability (IQ), epilepsy severity, cortical tuber count). Higher movement levels during movies were associated with higher parent-rated ADHD symptoms. Higher ADHD symptoms and actigraph-measured movement levels during movies were positively associated with ASD symptoms and negatively associated with IQ. Inter-individual variability of movement during movies was not associated with parent-rated hyperactivity or IQ but was negatively associated with ASD symptoms. There were no associations with tuber count or epilepsy. Our findings suggest that actigraph-measured movement provides a useful correlate of ADHD in TSC.

## 1. Introduction

Attention-deficit/hyperactivity disorder (ADHD) is a common neurodevelopmental condition characterised by developmentally inappropriate and impairing symptoms of inattention, hyperactivity, and impulsivity [1]. Excessive motor activity is an important aspect of ADHD. For example, longitudinal studies have shown that elevated motor activity levels in infancy and toddlerhood predict the presence and severity of both inattentive and hyperactive/impulsive symptoms later in childhood and adolescence [2,3]. Higher movement levels objectively measured with actigraphs (wearable sensors containing accelerometers that measure the intensity and frequency of body movements) are also consistently reported in children and adults diagnosed with ADHD compared to individuals without ADHD during both short (approximately 2 h) cognitive testing sessions and multiday recordings [4,5]. Multiday actigraph recordings in childhood have been used to predict later adolescent ADHD symptoms [6]. Further, excessive actigraph-measured motor activity during cognitive task performance differentiates individuals with persistent ADHD in adulthood from those with remitted ADHD and typically developing controls [7]. Due to the robustness of elevated objectively measured activity levels in ADHD and the sensitivity of this measure to differences in long-term diagnostic outcomes, it has been proposed that actigraphy could have value in aiding the clinical diagnosis of ADHD [8,9,10]. Recently developed products have incorporated actigraph technology in pursuit of this aim [11,12]. Actigraph-measured movement levels are also increasingly being included as outcome measures in clinical trials testing the efficacy of treatments for ADHD [8,9,13].

However, there are limited data using objectively measured activity to assess ADHD symptoms in more complex presentations of the disorder, such as in young people with developmental genetic disorders such as tuberous sclerosis (TSC). TSC is characterised by the growth of hamartomas in multiple organs, including the kidneys, lungs, heart, liver, and brain [14]. These lesions stem from genetic mutations in either the *TSC1* or *TSC2* genes, whose proteins form an essential complex for regulating development [15]. Brain pathology is extremely common in TSC and is estimated to occur in >90% of affected individuals [14]. In addition to disrupting brain structure, brain hamartomas (“cortical tubers”) and surrounding dysplastic tissue can serve as epileptogenic foci, leading to additional neurological problems. These developmental perturbances are further associated with intellectual disability (ID), autism spectrum disorder (ASD), and behavioural problems in TSC [16]. Epidemiological and clinical studies have reported ADHD symptom rates of 20–60% in TSC [17,18,19,20], which is considerably higher than the approximate 5% prevalence of ADHD in the general population [21].

There are a number of important reasons for investigating objective measures of activity levels in complex developmental disorders. First, actigraphs may be useful for diagnosis and treatment-monitoring of ADHD in these settings, as has been proposed in idiopathic ADHD [4,9]. Assessment of ADHD in conditions such as TSC is complicated by the presence of numerous comorbidities. For ID, especially, there are fewer rigorously tested tools available to measure ADHD symptoms than in idiopathic ADHD [22]. Objective activity sensors, especially minimally invasive ones such as actigraphs, may provide an additional method of measurement that could be useful in complex manifestations of ADHD. Second, in some complex developmental disorders, objective activity measurements can be modelled against known genetic and anatomical lesions that may cause ADHD by disrupting key neurodevelopmental processes. Importantly, aetiological factors revealed via this strategy may be relevant to idiopathic ADHD as well, as has been proposed for other psychiatric issues [23]. For example, research in TSC has revealed brain regions and epileptic signatures linked to the development of ASD [24,25]. While there are many proposed aetiological links between TSC pathologies and ADHD [26], few studies have investigated these associations, and findings so far have been variable. One study reported an association between subependymal nodules (SENs, hamartomas lining the ventricles) and ADHD in TSC [20], but other studies have not replicated this finding [27]. Another study reported an association between status epilepticus history and “behavioural symptoms” in TSC, but they did not differentiate between ADHD and ASD [28]. A third advantage of using actigraph-measured movement to investigate ADHD in individuals with TSC is that indices of intra-individual variability (IIV) can be derived even if individuals have co-occurring ID. Increased IIV is robustly associated with ADHD [29] and has been proposed to be a specific core impairment in the disorder [30,31]. IIV is typically measured as response-time variability (RTV) in speeded reaction time (RT) tasks. Such tasks are often not suitable for individuals with ID, either because the individuals are unable to follow the task instructions or due to motor coordination difficulties that frequently accompany ID [32]. Previous studies have indicated actigraph-measured movement IIV to be elevated in children with ADHD compared to controls [33,34,35]. Applying this approach in TSC presents an alternative opportunity to objectively investigate one of the proposed core neurocognitive mechanisms associated with ADHD in individuals in whom this would otherwise not be possible. Although there are limited data on actigraph use in TSC (let alone for ADHD monitoring), proof-of-concept data have successfully applied actigraphy to measure sleep-related issues in those with Fragile X syndrome [36].

In the current study, we aimed to investigate these potential benefits by firstly assessing whether objective, actigraph-measured movement levels and variability of movement (movement IIV) would be associated with parent-rated ADHD symptoms in a sample of young people with TSC and varying levels of intellectual ability, epilepsy, and symptoms of ASD. Actigraph measurements were obtained in two situational contexts, movie viewing and cognitive testing, during a short (approximately 1 h) daytime testing battery. Parent-rated ADHD symptoms were measured with the Strengths and Difficulties Questionnaire (SDQ) [37], which has been used for the assessment of ADHD in populations with ID [38,39]. We hypothesised that greater actigraph-measured movement and movement IIV would be associated with more severe parent-rated ADHD symptoms, which is consistent with findings in individuals with idiopathic ADHD and more homogeneous ability levels [4,5]. Second, we aimed to explore which features of TSC, including the number of cortical tubers and the severity of epilepsy, ID, and ASD, were associated with our actigraph and parent-rated indices of ADHD symptomatology. Due to the lack of consistent findings concerning associations between ADHD symptoms and features of TSC in previous work, we treated this second analysis as exploratory rather than hypothesis-driven.

## 2. Materials and Methods

### 2.1. Participants

This study utilised a subsample of the Tuberous Sclerosis 2000 (TS2000) study, a longitudinal investigation that has collected prospective data on a cohort of 125 individuals with TSC in the United Kingdom since 2005 [40]. Thirty individuals (11–21 years old) from the TS2000 cohort completed actigraph assessments in a laboratory visit, or, if they were unable to attend the visit due to medical or other complications, during a home visit. Data on ADHD and ASD symptoms, IQ, and epilepsy were collected at a separate assessment within 13 months of the actigraph testing session (mean = 5.8 months, SD = 5.21). Tuber count and genotype data were collected in earlier phases of the study (2001–2015). Ethical approval for this study was given by the NHS National Research Ethics Service (NHS Edgbaston REC 00/7/061). Parents gave written informed consent for participants younger than 16 years old. Participants aged 16 years and older provided written informed consent, unless ID prevented this, in which case parental written informed consent with the participant’s verbal assent were obtained.

### 2.2. Assessments

#### 2.2.1. Behaviour and Intellectual Ability

*ADHD symptoms:* Parent-rated ADHD symptoms were measured using the SDQ Hyperactivity/Inattention subscale, which consists of 5 questions assessing hyperactivity, impulsivity, and inattention [37]. Higher scores reflect greater ADHD symptoms.

*Intellectual ability:* IQ was measured with the two-subtest version of the Wechsler Abbreviated Scale of Intelligence–2nd Edition (WASI-II) [41], a standardised instrument used to assess intellectual ability, for all participants who could complete this assessment (*N* = 20). Age- and sex-normed full-scale intelligence quotients (FSIQ, normative mean = 100, SD = 15) were used in analysis. For participants who could not complete the WASI-II due to intellectual disability (*N* = 10), the Vineland Adaptive Behaviour Scales–2nd Edition (VABS-II) [42] was completed with participants’ parents/caregivers to assess adaptive functioning. The age- and sex-normed Adaptive Behaviour Composite (ABC, normative mean = 100, SD = 15) of the VABS-II was used to estimate IQ in these participants. The use of adaptive functioning to estimate intellectual ability in individuals who are unable to complete a standardised intelligence test is a well-established method in intellectual disability research and has been used in all previous analyses of the TS2000 Cohort [25,43,44].

*ASD symptoms:* Symptoms of ASD were assessed using the Autism Diagnostic Interview-Revised (ADI-R), a comprehensive, structured interview conducted with a parent/guardian of each participant [45]. The total ADI-R algorithm score was used in analyses. Higher scores reflect greater ASD symptoms.

#### 2.2.2. Features of TSC

*Genotype:* Determination of participants’ *TSC* mutations was carried out by either the East Anglian Medical Genetics Service (Cambridge) or the Institute of Medical Genetics (Cardiff). Multiplex ligation-dependent probe amplification was used to assay mutation status, with additional testing for whole exon deletions [40]. Participants for whom testing revealed no conclusive mutation were labelled as “no mutation identified” (NMI) in statistical analyses. Four participants did not receive genetic testing.

*Epilepsy:* The Early Childhood Epilepsy Severity Scale (E-Chess) was used to give a current index of epilepsy severity over the 3 months prior to testing [46]. The E-Chess contains 6 items assessing the frequency, type, and treatment of seizures. Total E-Chess scores were used in analyses, with higher scores indicating more severe epilepsy.

*Brain imaging:* Copies of clinical brain scans were procured from participants’ physicians. Clinical MRI or CT scans were assessed by two independent neuroradiologists, using a previously described system for coding the number and location (i.e., brain lobe) of cortical tubers and SENs [40,43]. In the current study, the total number or cortical tubers and SENs were used in analysis.

#### 2.2.3. Actigraphs

Participants wore two Move 3 actigraphs (movisens GmbH; Karlsruhe, Germany), one at their right hip and one on the wrist of their non-dominant hand (dominance assessed by asking which hand they use for writing, drawing, or similar tasks). Actigraph data were collected while participants completed two cognitive tests: the Test for Reception of Grammar (TROG-2) [47] and the WASI-II [41], and while they watched movie clips (a nature show and children’s cartoon). Actigraph measurements were collected for 22.2 min (SD = 6.7 min) of cognitive testing on average and for 28.1 min (SD = 6.2 min) of movie viewing on average. Movie clips were standardised for all but 4 participants, who would only tolerate a movie of their choosing while wearing the actigraphs (see Statistical Analysis Section 2.3 for details of how this was adjusted for in analysis).

Activity levels were initially computed in units of *mean movement* (average movement acceleration over 10-s intervals, averaged for the entire duration of the tasks or movies) and *threshold movement* (number of accelerations greater than 0.01 G per minute, sampled at 64 Hz, averaged for the entire duration of the tasks or movies). The coefficient of variation (CoV) (mean/standard deviation * 100) for these measurements was computed for each participant to provide a measure of IIV. As mean movement and threshold movement were highly correlated within both task and movie settings (data not shown, *R* > 0.8, *p* < 0.001), the two measures were standardised using z-scores and averaged to yield a single *actigraph-measured movement* measure during cognitive testing and another during movie viewing. CoVs were standardised and averaged in the same manner to yield an *actigraph-measured movement IIV* during movies and cognitive testing.

### 2.3. Statistical Analysis

All statistical analyses were conducted in Jamovi [48], with *α* = 0.05. To test the hypothesis that actigraph-measured movement and movement IIV would be positively associated with parent-rated ADHD symptoms, multiple linear regression models were constructed with actigraph variables (actigraph-measured movement and actigraph-measured movement IIV) as predictors and SDQ Hyperactivity/Inattention scores as the outcome; separate models were used for actigraph-measured movement and for actigraph-measured movement IIV. Age and testing location (home versus lab) were controlled for in these models in a second block. In addition, to check whether the fact that four participants would only tolerate a movie of their choosing influenced the findings, regression models examining the prediction of SDQ Hyperactivity/Inattention from actigraph variables during movie viewing were repeated with these participants omitted.

Next, to explore which features of TSC would be associated with actigraph measurements and parent-rated ADHD symptoms, Spearman correlations were computed between actigraph measures and SDQ Hyperactivity/Inattention and the variables IQ, ASD symptom scores, tuber counts, SENs, and epilepsy severity scores. Partial correlations were used to include adjustment for age and testing location. Due to the small number of participants with *TSC1* mutations (*n* = 3), associations between TSC genotype and actigraph measures and parent-rated ADHD symptoms were not examined.

## 3. Results

### 3.1. Sample Characteristics

Thirty individuals with TSC were assessed with actigraphs; characteristics of the sample and assessments are presented in Table 1. Participants were mostly female (20 females, 10 males), with an average age of 15.3 years. Most participants (27/30) were 11–18 years old; three were between 19–21 years old. Mutations in *TSC2* were the most common mutation type (18/30). The average IQ of the sample was 70.6, denoting borderline intellectual impairment. Individuals ranged from having severe IQ deficits to having a normal IQ (range = 35–108); 14 participants had an IQ < 70. Due to issues of tolerability and intellectual impairment, some participants did not complete all assessments (see Table 1). More participants completed actigraph testing with movies (*n* = 28) than with cognitive testing (*n* = 20).

### 3.2. Associations between Actigraphy and Parent-Rated ADHD Symptoms

During cognitive testing, actigraph-measured movement was not associated with SDQ Hyperactivity/Inattention before (*R*^2^ = 0.106, adj. *R*^2^ = 0.056, *F* = 2.1, *p* = 0.161) or after (*R*^2^ = 0.127, adj. *R*^2^ = 0.0, *F* = 0.8, *p* = 0.523) adjustment for age and testing site (Figure 1A). Similarly, actigraph-measured movement IIV during cognitive testing was also not associated with ADHD before (*R*^2^ = 0.131, adj. *R*^2^ = 0.083, *F* = 2.7, *p* = 0.116) or after adjustment for age and testing site (*R*^2^ = 0.181, adj. *R*^2^ = 0.028, *F* = 1.9, *p* = 0.349) (Figure 1B).

Actigraph-measured movement during movies, on the other hand, was positively associated with ADHD symptoms (*R*^2^ = 0.310, adj. *R*^2^ = 0.276, *F* = 9.0, *p* = 0.007; *actigraph-measured movement during movies:* β = 0.557, *p* = 0.007) (Figure 1C). After adjustment for age and testing site, the overall model became marginal, yet the proportion of explained variance remained large (*R*^2^ = 0.321, adj. *R*^2^ = 0.208, *F* = 2.8, *p* = 0.067) and actigraph-measured movement remained a significant predictor (β = 0.566 *p* = 0.011). Movement IIV during movies trended towards negatively predicting ADHD symptom ratings (*R*^2^ = 0.172, adj. *R*^2^ = 0.132, *F* = 4.2, *p* = 0.054; *actigraph-measured movement IIV during movies:* β = −0.417 *p* = 0.054) (Figure 1D), yet this association became non-significant after adjustment for age and testing site (*R*^2^ = 0.179, adj. *R*^2^ = 0.042, *F* = 1.3, *p* = 0.304; *actigraph movie IIV*: β = −0.413 *p* = 0.075).

The pattern of results were similar when omitting participants who only tolerated movies of their choosing; actigraph-measured movement during movies was significantly positively associated with parent-rated ADHD symptoms (*R*^2^ = 0.333, adj. *R*^2^ = 0.296, *F* = 9.0, *p* = 0.008; *actigraph-measured movement during movies:* β = 0.577, *p* = 0.008), and this association became marginal after adjusting for age and testing location (*R*^2^ = 0.353, adj. *R*^2^ = 0.232, *F* = 2.9, *p* = 0.067; *actigraph-measured movement during movies:* β = 0.585, *p* = 0.013). Actigraph-measured movement IIV was also significantly negatively associated with ADHD symptoms (*R*^2^ = 0.257, adj. *R*^2^ = 0.216, *F* = 6.2, *p* = 0.022; *actigraph-measured movement IIV during movies:* β = −0.507, *p* = 0.022), but this association became non-significant after adjustment for age and testing site (*R*^2^ = 0.275, adj. *R*^2^ = 0.139, *F* = 2.0, *p* = 0.152; *actigraph-measured movement IIV during movies:* β = −0.499, *p* = 0.037).

### 3.3. Associations between Clinical Features of TSC, Parent-Rated ADHD Symptoms, and Actigraph Measures

Spearman correlation coefficients were computed between clinical feature of TSC (IQ, epilepsy, ASD symptoms, tuber count, and SENs) and parent-rated ADHD symptoms and actigraph measures. Actigraph measurements during cognitive testing were not included in this analysis, as these variables showed no association with parent-rated ADHD symptoms. IQ was significantly negatively correlated with both parent-rated ADHD scores (*rho* = −0.597, *p* = 0.002) and actigraph-measured movement during movies (*rho* = −0.643, *p* < 0.001). These associations were maintained after partialling out the effects of age and testing site (respectively: *rho* = −0.534, *p* = 0.003; *rho* = −0.656, *p* < 0.001) (Figure 2A,B). IQ was also significantly positively associated with actigraph-measured movement IIV during movies (*rho* = 0.491, *p* = 0.008), but this association became marginal after adjustment for age and testing site (*rho* = 0.363, *p* = 0.057) (Figure 2C).

Parent-rated ADHD symptoms tended to be positively correlated with ASD symptoms (*rho* = 0.403, *p* = 0.070), and this association became significant after adjustment for age (*rho* = 0.451, *p* = 0.040) (Figure 2D). Actigraph-measured movement during movies was also significantly positively associated with ASD symptoms on the ADI-R (*rho* = 0.704, *p* = < 0.001), and this relationship remained significant after adjustment for age and testing location (*rho* = 0.649, *p* < 0.001) (Figure 2E). Actigraph-measured movement IIV during movies was significantly negatively associated with ASD symptoms, both before (*rho* = −0.738, *p* < 0.001) and after (*rho* = −0.684, *p* < 0.001) adjustment for testing location and age (Figure 2F).

Neither current epilepsy severity, total tubers, nor SENs were correlated with movie activity, movie activity IIV, or parent-rated ADHD symptoms (all *rho* < 0.190, *p* > 0.05).

## 4. Discussion

This study aimed to expand previous research on actigraph use for the assessment of ADHD by applying this tool in a cohort of individuals with TSC. Previous studies using actigraphy to assess ADHD have shown robust associations between ADHD and objectively measured movement, though they have focused on idiopathic development and individuals with normal or high IQ [4,5]. In this study, actigraph-measured movement during movie viewing was positively associated with parent-rated ADHD symptoms in a symptomatically diverse sample. Adjustment for age and location of testing reduced the significance of this association in our model; however, the magnitude of actigraph-measured activity as a predictor remained consistent. These results provide initial support for the notion that the association between ADHD and objective movement is maintained in settings with more heterogenous profiles of intelligence and behavioural issues.

While actigraphy has not been used to study ADHD in TSC, other objective instruments including continuous performance tasks (CPTs) have, although these are only suitable for individuals with higher intellectual abilities. One study by Vries et al. [49] reported a lack of correlation between parental behavioural ratings and objectively measured attention deficits on a CPT. The authors proposed that this lack of agreement could indicate “silent” attentional deficits that are unnoticed by parents. However, an alternative explanation could be that behavioural rating scales for neurotypical individuals, such as the Conners Parent Rating Scale [50] used by de Vries et al. [49], do not accurately index ADHD symptoms in populations with ID or ASD, as has been suggested elsewhere [22]. For this reason, the SDQ was used for parental ratings of ADHD symptoms in this study, as this scale has more established use in populations with ID [37,38,39]. Our findings indicate that actigraph measures may provide a useful alternative to CPTs for assessing ADHD-related difficulties in TSC.

We found associations between actigraph-measured movement and parent-rated ADHD symptoms during movie viewing, but not during cognitive testing. This dissimilarity could stem from context-dependent differences in the presentation of hyperactive ADHD symptoms [51,52]. Situational effects on movement, such as the type of school activity performed or classroom setting, have been documented in previous actigraph studies in idiopathic ADHD [33,53,54]. The cognitive tests used during actigraphy in this study required participants to interact with investigators and respond to questions in a one-on-one structured session; these conditions could have the effect of stymieing hyperactivity [33]. Previous actigraph studies have indicated that ADHD-related hyperactivity is modulated by situational demands on executive functioning, with more intensive situations eliciting more hyperactivity [55,56,57]. It is unclear if this same pattern holds true for individuals with TSC or other disorders who may have significant cognitive issues due to ADHD as well as other psychological and neurological issues. It is possible that the two actigraph contexts in our study had differential executive functioning burdens (even if less intense than tasks designed to probe executive functioning), causing differential effects on hyperactive motor behavior. The discrepancy between recording contexts in this study may also relate to autonomic dysregulation in ADHD. Hyperactivity and impulsivity in ADHD are possible manifestations of a regulatory mechanism to boost arousal in an otherwise hypo-aroused state [58,59]. The lack of increased actigraph-measured movement during cognitive testing may correspond with increased arousal stemming from the task itself. However, we did not capture any autonomic data during actigraph testing to support or refute this notion. It should also be noted that a larger number of participants completed actigraph testing with movies than with cognitive testing in the present study, so the results could also relate to differential power.

Results further indicated IQ and ASD symptoms to be correlates of actigraph-measured movement in TSC. Participants with lower IQs exhibited higher movement levels during movies as well as more ADHD symptoms. This finding resonates with previous cross-sectional studies that documented a higher prevalence of ADHD or its symptoms (e.g., hyperactivity) in children with ID [17] or lower intelligence [20], as well as studies of attentional deficits in those with TSC [49]. ASD symptoms similarly were associated with higher movement levels during movies and higher ADHD symptom scores. This could indicate the presence of motor perturbances that are common to both ASD and ADHD, which is a notion supported by previous work outside of TSC [60,61,62]. For this reason, further studies of TSC (or other developmental disorders) should note that actigraphy is likely to capture motor patterns related to both ADHD and ASD.

We found weak evidence that participants with higher ADHD symptom ratings had lower actigraph-measured movement IIV. We also found that movement IIV was negatively associated with ASD symptoms, and movement IIV also trended towards being positively associated with IQ. Interestingly, the direction of these correlations was opposite to that of actigraph measured-movement; i.e., actigraph-measured movement being positively correlated with ADHD and ASD symptoms but movement IIV being negatively correlated with the two symptom measures. We had expected elevated movement IIV to be associated with more severe ADHD symptoms, given previous results indicating behavioural variability to be a hallmark of ADHD [29,31] as well as earlier actigraph studies showing increased movement variability in those with ADHD compared to controls during short (approximately 2–4 h) daytime classroom cognitive testing sessions [33,34,35]. The present data instead suggested that higher movement IIV was correlated with less movement, perhaps indicating that more hyperactive individuals were consistently moving at higher levels than less active ones. Although unexpected, this result does align with previous findings in an actigraph study from Tsujii et al. [54]. These authors reported lower movement variability of children with ADHD compared to children with PDD/hyperactivity and typically developing controls during a school recess setting, as well lower IIV in all groups during recess compared to the classroom. The reason for the discrepancy across studies is unclear, although it could be an artefact of the differences in methods of data extraction from actigraphs between studies. Similar methods of deriving movement levels from actigraphs have been used in previous ADHD actigraph studies, although choices regarding the epoch length, sampling frequency, and movement thresholds have varied across studies. The development of guidelines for actigraphy in ADHD research might be needed to inform future designs; such guidelines have been proposed for physical activity research with actigraphs [63,64]. The pattern of actigraph-measured movement IIV in our study might also relate to the differences in movements exhibited by those with stronger or milder ADHD symptoms. That is, those with higher ADHD symptoms may have had consistently higher levels of activity, without much variability in the extent to which activity levels fluctuated from high to low, resulting in lower actigraph-measured movement IIV.

This study did not find evidence of associations between biological factors of TSC (i.e., epilepsy severity, tuber count, or SEN count) and actigraph measurements. Previous investigations into the biological correlates of ADHD in TSC have been limited and inconclusive. While one epidemiological study reported a correlation between SENs and ADHD symptoms [20], another larger study did not find this association [27]; the current study also did not find any association. *TSC2* mutations are associated with more severe pathology in general [14], but they have not been specifically linked to ADHD. The development of psychopathology in TSC is complicated by the interrelatedness of cortical tubers, epileptic issues, and intellectual impairment. More advanced methods, such as structural equation modelling, have been used in the full TS2000 sample to model the developmental origin of IQ deficits in TSC [43]; similar methods in larger samples might be needed to identify the factors most important for hyperactivity or other ADHD symptoms in TSC.

There are some limitations to this study that should be noted. There was variability in the manner by which participants underwent actigraph assessment, e.g., testing was done in two locations (home and lab), and participants did not all see the same movie clips, as some children would only tolerate movies of their choosing. We addressed these issues by statistically adjusting for the testing location and rerunning analyses with some participants excluded, which did not alter the associations with ADHD symptoms. We also did not control for the time of day when actigraph assessments were collected, which is known to affect the presentation of ADHD [65,66] and has been shown to affect actigraph assessment [67]. Unfortunately, this was not an option as participant families were either being visited in or were travelling from multiple locations across the UK, and many participants had impairing behavioural and physical symptoms that prevent strict scheduling. These issues also prevented the use of longer actigraph recordings, which might be useful for capturing more natural and subtle patterns of hyperactivity [6,10,35]. There are also other major factors that contribute to hyperactivity in both idiopathic ADHD and otherwise, such as sleep disorders, which we were not able to assess in the present study. Additionally, actigraph use is totally novel in TSC, and more work needs to be done to assess the validity and reliability of actigraph data in this population. Moreover, the present analysis comes from a small sample; for these reasons, results from this study need to be taken cautiously and replicated.

## 5. Conclusions

This preliminary study found that actigraph-measured movement was significantly associated with parent-rated ADHD symptoms in a sample of young people with TSC and varying levels of ADHD symptoms as well as intellectual ability and autism symptoms. Actigraphy provided an objective measure of hyperactivity that was appropriate for individuals with diverse abilities and difficulties. Further development of the actigraph method in TSC may be useful for the study and clinical evaluation of ADHD in complex neurodevelopmental conditions.

## Figures and Tables

**Figure 1 brainsci-10-00491-f001:**
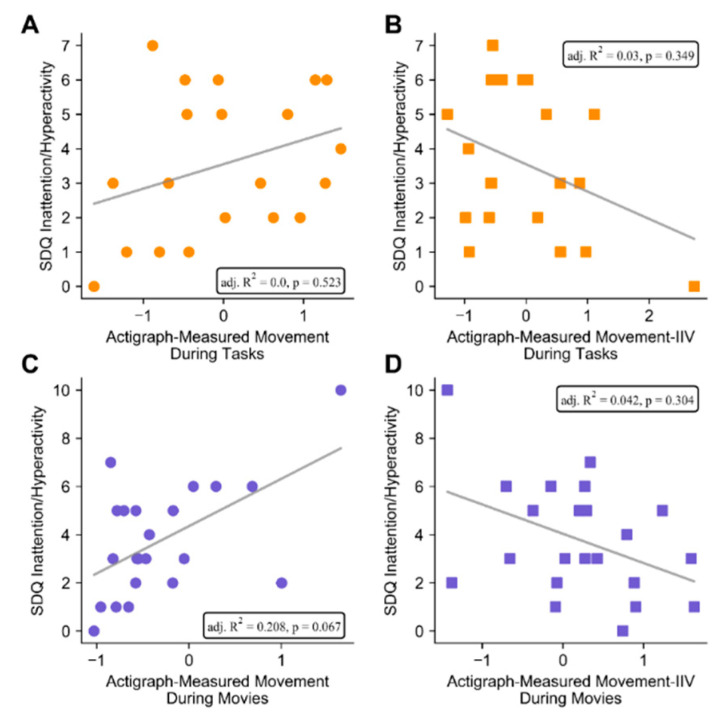
Associations between actigraph measurements and parent-rated attention-deficit/hyperactivity disorder (ADHD) symptoms. During cognitive testing, neither actigraph-measured movement (**A**) nor actigraph-measured movement IIV (**B**) were associated with parent-rated ADHD symptoms. Actigraph-measured movement during movies was positively associated with ADHD symptoms, though this association became marginal after adjusting for age and testing site (**C**). Actigraph-measured movement IIV trended towards being negatively associated with ADHD symptoms, but this association was non-significant after adjustment for age and testing site (**D**). Lines represent least-squares fit, while insets show adjusted *R*^2^ and *p* values for multiple regressions after adjustment for age and testing site.

**Figure 2 brainsci-10-00491-f002:**
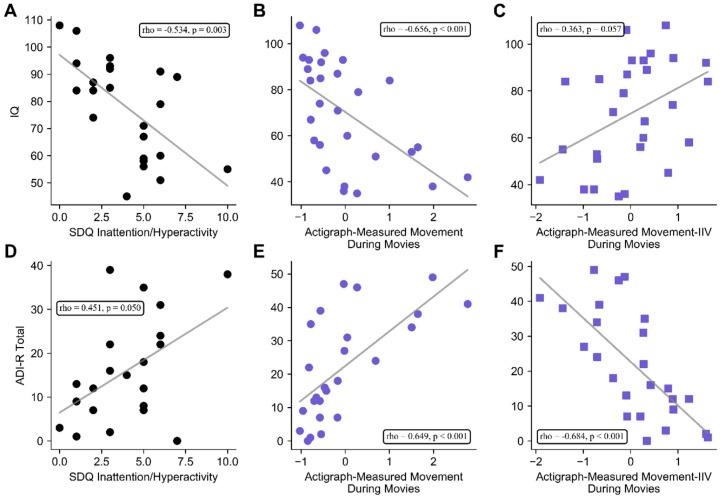
Associations between intellectual ability (IQ), ASD symptoms, and actigraph measurements. IQ was negatively associated with parent-rated ADHD symptoms (**A**) and actigraph-measured movement during movies (**B**); IQ also trended towards being positively associated with actigraph-measured movement IIV during movies (**C**). ASD symptoms were positively associated with ADHD symptoms (**D**) and actigraph-measured movement during movies (**E**). ASD symptoms were also positively correlated with actigraph-measured movement IIV during movies (**F**). Lines represent least-squares fit, insets show *rho* and *p* values for Spearman correlations after adjustment for age and testing site.

**Table 1 brainsci-10-00491-t001:** Sample characteristics and descriptive statistics for all assessments. NMI = no mutation identified, IIV = inter-individual variability, CoV = coefficient of variation, WASI-II = Wechsler Abbreviated Scale of Intelligence II, VABS = Vineland Adaptive Behavior Scale II, SDQ = Strengths and Difficulties Questionnaire, ADI-R = Autism Diagnostic Interview Revised, E-Chess = Early Childhood Epilepsy Severity Scale, SENs = subependymal nodules. * SEN data only available for 24 participants. Note that actigraph measurements were standardised using z-scores and so the means are zero and the SDs are close to 1.

Measure	Mean (Standard Deviation)	Participants (% of Total)
**Age**	**15.3 (3.4)**	**30 (100)**
**Gender**	**-**	**30 (100)**
male	-	10 (33.3)
female	-	20 (66.7)
**Mutation**	**-**	**26 (86.7)**
TSC1	-	3 (11.5)
TSC2	-	18 (69.2)
NMI	-	5 (19.2)
not tested	-	4 (13.3)
**Testing Site**	**-**	**30 (100)**
lab	-	16 (53.3)
home	-	14 (46.7)
**Actigraphy**	**-**	**30 (100)**
cognitive test movement	0 (0.957)	20 (66.7)
cognitive test movement IIV	0 (0.953)	20 (66.7)
movie movement	0 (0.972)	28 (93.3)
movie movement IIV	0 (0.880)	28 (93.3)
**IQ (WASI-II or VABS-II)**	**70.6 (22.4)**	**30 (100)**
**ADHD Symptoms (SDQ** **Hyperactivity/Inattention)**	**3.9 (2.3)**	**25 (83.3)**
**Autistic Symptom Severity** **(ADI-R)**	**21.4 (15.4)**	**27 (90.0)**
**Brain Lesions**	**-**	**28 (93.3)**
total tubers	19.3 (14.8)	-
frontal tubers	10.2 (8.2)	-
parietal tubers	3.2 (2.4)	-
temporal tubers	2.9 (3.1)	-
occipital tubers	3.9 (4.0)	-
cerebellum tubers	0.0 (0.0)	-
SENs *	4.0 (3.7)	-
**Epilepsy Symptoms (E-Chess)**	**6.0 (4.9)**	**26 (86.7)**

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
