# Peer review of "Actigraph-Measured Movement Correlates of Attention-Deficit/Hyperactivity Disorder (ADHD) Symptoms in Young People with Tuberous Sclerosis Complex (TSC) with and without Intellectual Disability and Autism Spectrum Disorder (ASD)"

_brainsci, 2020, doi:10.3390/brainsci10080491_

Round 1
Reviewer 1 Report
Dear authors
I appreciate the idea and novelty of the measurement, but you should underline and explain better the measurement used:
you did not describe accurately the methods, it is not clear how long the children wore the actigraphic recording (13 months or few hours?), is it recorded in the same time of the day for all children? Timing is important, since movements are also influenced by circadian factor. The authors did not investigate sleep disorders which is strong mediator factor for increased movements, and should be added to the limits of the study. The authors cited studies on ADHD but they did not specify if results are referred to daytime, 24 hours, sleep time.
Minor concerns:
Abstarct:
What is the difference in prediction and correlation?
First paragraph of introduction: specify when it is increased the motor activity detected by actigraphic recording (dytime, sleep time, 24 hours).
Second paragraph: “An important limitation of this work”: please change with there are limited data on…
Reviewer 2 Report
Dear authors, thanks for submitting your manuscript to Brain Sciences.
The study is interesting and present interesting findings which are relevant for our research field. The paper is very well written, and logically organised, it was a pleasure to read it and I think it will be a very valuable contribution to the journal. I have some comments, which I would like you to address, before accepting it for publication.
-I agree with the authors in saying that activity/movement profiles should be further analysed in ADHD, especially in relation with other factors (such as age or IQ, and co-occurring symptoms); it is good that your paper supports this idea!
-It is interesting that you found specific associations between actigraphy measures and ADHD symptoms, but only in one specific condition (i.e., when watching a movie). In relation to the paragraph at lines 301-309, do the authors thing that this finding might be somewhat associated with dysregulated physiological arousal state in ADHD, especially in more passive situations (see https://doi.org/10.1016/j.neubiorev.2019.11.001)? What do the authors think? If the authors share this idea, could they add some comments about it in the discussion? Increased movement level might in fact be a strategy to up-regulate (hypo-)arousal states during more passive and less engaging contexts, in ADHD.
-In relation to your findings about movement-IIV (M-IIV), could it be that you did not find any associations between ADHD and M-IIV, because those with higher ADHD symptoms showed overall increased movement activity, but never in the range of movement activity found in those with less ADHD symptoms (= less variability in movement activity)?
Thanks again for the opportunity of reading this manuscript and asking me to provide feedback on it.
Best wishes
